# Emergency department care experiences among youth with mental health concerns

**Laura K. Wells** [1], **Susan A. Bartels**[2,3], **Tania Nicholls**[2], **Melanie Walker** [2,3]*

**1** Department of Family Medicine, Queen's University, Kingston, Ontario, Canada, **2** Department of Emergency Medicine, Queen's University, Kingston, Ontario, Canada, **3** Department of Public Health Sciences, Queen's University, Kingston, Ontario, Canada

* melanie.walker@kingstonhsc.ca

## Abstract

Emergency departments (EDs) are important for mental health (MH) care among youth, however, are often ill equipped to meet their needs, resulting in care dissatisfaction. The objective of this research is to better understand the ED care experiences among youth with MH concerns. Data was used from a cross-sectional, mixed-methods study comparing ED care experiences among individuals who identify as equity-deserving with those who do not. Equity deserving groups are defined as individuals who face shared barriers to participating in society and care as a result of identifying with a certain community. For this sub-group analysis, participants who identified as youth aged 16–24 with a MH concern were compared to age-matched controls. Descriptive and inferential statistics, including chi squared tests were used to evaluate differences in sociodemographic and ED visit data between groups. Qualitative micronarratives were thematically analyzed to contextualize quantitative findings. A total of 81 youth with MH concerns and 108 youth controls were included in the analysis. Compared to controls, youth with MH concerns experienced greater: negative effects of their identity on ED experiences; disrespect in the ED; and negative feelings throughout their ED visit (p<0.001 for all). Themes identified in the qualitative analysis supported these findings including judgement and stigmatization, unmet care needs, lack of MH expertise and community connections, and negative impacts of the ED environment on MH experiences. This study adds to evidence demonstrating that youth with MH concerns have largely negative experiences with ED care in comparison to age-matched control group. Interventions to improve care could include improving MH expertise in the ED with existing or new staff and enhancing connections to community MH resources.

## Introduction

### Background

Approximately one in five Canadians will experience a mental illness in their lifetime, and many experience this for the first time during their youth [1]. Increasing prevalence of mental health (MH) concerns, coupled with reduced stigma and increased acceptance for MH

**Data Availability Statement:** Quantitative and qualitative data was used for the analysis in this study. The quantitative data reported in the manuscript is publicly available as supplementary

information as indicated in the manuscript. It is included as supplementary material titled "S1_Dataset" and is referenced in the manuscript text. The qualitative data reported in the manuscript is not made publicly available due to ethical concerns regarding participant privacy, the inclusion of structurally disadvantaged small groups, and the possibility of re-identification of participants when multiple datapoints are viewed together with the narrative transcripts. Excerpts of transcripts relevant to the study can be made available upon request by contacting the corresponding author, MW, or Jessica Montagner (Jessica.Montagner@kingstonhsc.ca), Research Manager in the Department of Emergency Medicine at Queen's University.

**Funding:** This secondary analysis was an unfunded student project. The primary author (LW) did not receive any direct funding for this sub analysis. The funders of the parent research study from which data was used for this work include the Queen's University Catalyst Fund (PI S. Bartels), Clinical Teachers' Association of Queen's University (PI S. Douglas), Queen's University Fund for Scholarly Research and Creative Work and Professional Development (PI M. Walker). This research was also supported, in part, thanks to funding from the Canada Research Chairs Program. For this sub analysis the funders had no role in study design, data collection and analysis, decision to publish, or preparation of this manuscript.

**Competing interests:** The authors have declared that no competing interests exist.

concerns, leads to a growing demand for MH care for youth in Canada [1]. In Canada, there has been a rising demand for youth mental health (MH) care in emergency departments (EDs), with a doubling of MH-related ED visits among Canadian youth aged 14–24 between 2009–2017 [2]. The increasing prevalence of MH issues has been exacerbated by the COVID-19 pandemic, with youth experiencing worsening depression, anxiety, psychological stress, suicidal ideation, and decreased psychological wellbeing [3–5].

For youth with MH concerns (YMHC), the ED can act as a first point of MH care contact as it can temporarily overcome barriers to accessing timely outpatient social and MH services in the community and does not require referrals or appointments to access [1, 2, 4, 6]. However, busy ED settings are often not resourced to respond to MH presentations, meet the complex needs of youth, or facilitate continuity of care after visiting the ED [4, 7]. This can lead to extended ED wait times and dissatisfaction with care services [4]. Given the role of EDs in providing MH care among youth, it is essential they are set up with appropriate resources and expertise to support YMHC [6].

Individuals with a MH concern, including YMHC, often face barriers to care and considered an equity-deserving group (EDG) [8]. An EDG is a group of individuals that face barriers to accessing opportunities and resources in society, and experience systemic discrimination as a result of a particular identity [9]. Equity and MH often intersect, with health inequities influencing individual's MH, and having a MH concern presenting inequities in itself [8]. It is therefore important to understand and improve YMHC experiences and interactions with the health system, including EDs, to reduce health and social inequities for this population.

Evidence among adults with MH concerns suggest largely negative ED experiences, but perspectives among YMHC are lacking [10–16]. The limited research among youth suggests concerns such as lack of privacy in the ED; long wait-times contributing to worsening of MH conditions; and negative experiences with health care provider attitudes and behaviour [17–19]. Existing quantitative and mixed-methods research among youth under the age of 18 in Canadian pediatric EDs report satisfaction with care when there is connection to community Emergency Medicine resources, confidentiality and respect in the ED, and positive provider attitudes and interpersonal skills. In contrast, YMHCs experience care dissatisfaction with respect to MH symptom relief, wait-times, the impacts of COVID-19, and accessing MH care in the community [20, 21]. However, these studies, along with most of the existing literature for adults, are primarily qualitative in nature, limiting generalizability of findings [11–13, 16–19] and lack control groups, thus preventing the identification of experiences unique to YMHC [10, 11, 15, 17–21]. There is therefore a need for further research among YMHC that contributes quantitative findings that support the qualitative literature. Furthermore, it would be important to identify ED care experiences unique to youth with MH concerns as compared to youth without MH concerns to identify targeted interventions to improve the care of this EDG.

## Purpose

This research aims to better understand how YMHC experience ED care, and how those experiences differ from youth without MH concerns.

## Materials and methods

### Study design and data source

This research used a sub-set of data from a parent study to examine the first-person ED experiences among youth. Data was used from a large cross-sectional, mixed methods study examining ED care experiences among equity-deserving groups (EDGs) in Kingston, Ontario,

Canada including YMHC [22]. The parent study compared the care experiences of those who identified as member of an EDG to a control group of individuals who did not identify as equity-deserving [22]. Equity deserving groups are defined as individuals who face shared barriers to participating in society and care as a result of identifying with a certain community. Data for the parent study were collected using Spryng.io [23], a 'sensemaking' survey software that captures both quantitative data and qualitative lived experiences. Sense-making is described as an ethnographic approach to research whereby participants can self-interpret their own responses [24]. This approach allows for the collection of a large number of experiences along with self-interpretation of these experiences to contextualize responses [22–24].

Participants for the parent study were recruited from June to August 2021 from the two EDs in Kingston, the Kingston Health Sciences Centre's ED and Urgent Care Centre, and approximately 20 community partner organizations in Kingston. Using convenience sampling during hours of research assistant availability (Monday to Friday, 9am-9pm), all eligible individuals were invited to participate at the Kingston Health Sciences Centre's ED and Urgent Care Centre, and community partner organizations by a trained research assistant. To participate, individuals had to be aged ≥16 years, proficient in English, and visited a Kingston ED within the previous two years. All individuals who met this criteria were able to participate, regardless of their identification with an EDG or not. Individuals did not have to present to the ED for a MH concern, but rather any ED care experience could be shared.

The initial survey in the parent study was created with input and collaboration from individuals and community organizations who identified as members of various EDGs [22]. The survey took approximately 15 minutes to complete, and was completed via electronic tablet by the participant. The survey first asked participants to share a recent (within 24 months) positive or negative experience about themselves or someone they accompanied to the Kingston Health Sciences Centre's ED or Urgent Care Centre via audio-recording or typing free-text. The question was phrased in a neutral way in that the ED experience shared did not have to be relating to a MH concern nor part of an equity-deserving identity. Participants then responded to a series of interpretation questions (see S1 Table) that involved plotting perspectives about the shared ED experience between three options (triads) or two options (dyads), an example of which is provided in S1 Fig. To limit social desirability bias, all survey questions were framed in a neutral way such that there was no option within a given question that could be easily perceived as being more 'correct' than another. Participants then completed a series of multiple-choice questions about their ED visit to further contextualize responses and sociodemographic characteristics (see S2 Table). One multiple-choice question asked participants to self-identify as a member of the listed EDGs in the question, and could choose to identify with up to 3 different EDGs (see S2 Fig). Participants were able to respond with 'not sure' or 'prefer not to say' for any question.

**Selection of participants.**   A sub-set of data from the parent study was used for this research to examine the first-person ED experiences among youth who participated (i.e. youth who shared a story about themselves) (see S1 Dataset). Youth was defined as individuals who self-identified in the survey as aged 16–24 years. YMHC were classified as all youth who self-identified in the survey as belonging to the EDG of having a MH concern (S2 Fig). Youth did not have to share a story relating to having a MH concern to be classified as a YMHC. A suitable age-matched control group was selected by including all youth who self-identified as not belonging to any EDG (S2 Fig). This was to ensure controls were correctly classified as not having a MH concern, given: 1) Participants could only identify with up to 3 EDG in the survey, and therefore may have identified as having a MH concern but not been able to indicate it, and; 2) Participants may have interpreted other EDG, such as having a disability or using alcohol or other substances, as having a MH concern.

**Primary outcome.** The primary outcome of this study was to identify how YMHC experienced ED care, and if those experiences differed compared to youth without MH concerns.

### Data analysis

**Statistical analysis.** Descriptive and inferential statistics using chi-squared tests were used to evaluate sociodemographic and ED visit data between YMHC and age-matched controls (SPSS Statistics Premium Campus Edition v28). P-values $< 0.05$ were considered statistically significant. Descriptive and inferential statistics did not include missing data, and responses of 'not sure' or 'prefer not to say' were recoded as missing data for the purposes of the analysis.

The data from triad and dyad questions from Spryng.io were exported to Tableau Desktop v2022.1 [25] and visually examined for patterns. Triad data were analyzed using R scripts (R v3.4.0) [26] to generate geometric means and 95% confidence ellipses represented visually around each group mean [27]. Confidence ellipses that did not overlap represented statistically significant differences between groups. Dyad data were analyzed using Kruskal-Wallis H statistical tests to evaluate between-group differences (IBM SPSS Statistics v24.0.0.0) [28]. Results of statistically significant triad and dyad analyses were selected and presented along with the results of the thematic analysis to contextualize findings.

**Qualitative analysis.** The qualitative component of this research was completed in accordance with the COREQ (Consolidated criteria for Reporting Qualitative research Checklist) (see S3 Table). Qualitative data was derived from the short narratives shared by participants in the survey either via free-text or audio recording, termed 'micronarratives', describing ED care experiences among YMHC. The qualitative shared experiences from participants were briefer than traditional qualitative transcripts and hence termed micronarratives. Any audio recordings were transcribed verbatim and exported to a Microsoft Excel workbook. The micronarratives were thematically analyzed using an inductive and deductive coding approach [29]. A master codebook, created by study personnel based on the parent survey and existing evidence, was used as an initial framework. Any newly identified codes from the qualitative review were added to the master codebook. All micronarratives for YMHC were coded by one author (L.W.) and 20% were double coded (by M.W.) as a validity check. The codes were then grouped thematically and agreed upon by team consensus [29].

The thematic findings of this analysis were subsequently member-checked through a community-based participatory approach to knowledge sharing [30]. In-person community a focus group discussion (FGD) that included members of the research team, local service providers and individuals who identified as having a MH concern, to situate the findings for accuracy and discuss meaningful quality improvement strategies [30]. FGD were transcribed and de-identified, and subsequently reviewed by L.W. to integrate and generate themes based on the discussion. Lastly, experts in the field of youth MH reviewed this paper to assess and situate the accuracy of these findings and validate subsequent suggestions for change.

**Situating the researchers.** The primary author, L.W., was a senior medical student at time of data collection and initial manuscript writing; is a current resident physician training in Family Medicine and Public Health and Preventive Medicine; was a lead research assistant for data collection during the parent study, and a 'millennial'. These roles and identities allow her to situate participant responses in the context of her experiences in the healthcare system, interactions with participants, and as a young adult. Authors S.B. and M.W. were primary investigators on the parent study and are research scientists in Emergency Medicine, global health, and health equity. Author S.B. is also a practicing Emergency Medicine physician and thus has expert knowledge of the ED environment, culture, and constraints. Author T.N. has a Master of Social Work and works as a Social Worker in the MH section of the KHSC ED in

addition to work experience as a counsellor with youth in an outpatient setting. Authors L.W., S.B., & M.W. contributed to a literature review on ED care experiences for people with MH concerns. All authors therefore have working knowledge of barriers and facilitators to MH care.

**Ethical considerations.** The parent study was approved by the Queen's University Health Sciences and Affiliated Teaching Hospitals Research Ethics Board (protocol #6029400). Participants provided written informed consent by ticking a box on the electronic survey and were offered a $5 coffee card for completing the survey. No identifying information was collected.

## Results

### Participant characteristics

A total of 1973 unique participants shared 2114 experiences about ED care in the parent study. Of those, 316 participants identified as youth aged 16–24 and shared a first-person experience, among which 81 identified as having a MH concern, and 108 did not identify with any EDG (i.e. 'controls'), creating a sample of 189 youth for this analysis (see Fig 1).

The parent study had a response rate of 41.6%, with 2579 individuals declining to participate in the study because of not having a previous visit to the ED within the last 24 months (46.9%), not being interested in participating (18.8%), feeling too unwell to take part (14.7%), and already having participated (6.5%).

A total of 81 micronarratives among YMHC were thematically analyzed. One FGD was included for thematic analysis which was held among community members who self-identified as having a MH concern and/or as a person who use substances, and included 17 participants, 11 men, and 6 women.

Compared with controls, YMHC identified more frequently as women, non-binary, and gender diverse, and more often struggled to make ends meet, with p<0.001 for all (Table 1). The majority of YMHC identified with more than one EDG, with 32.1% identifying with more than 1 EDG and 50.6% with more than 2 EDG (Table 1). This including identifying as having a disability (33.3%), as LGBTQ2S+ (29.6%), as using alcohol or other substances (28.4%), as being vulnerably housed (9.9%), and/or as Indigenous (8.6%).

### ED visit characteristics

Compared with controls, YMHC reported: more frequent visits to the ED in the last two years prior to survey completion (p<0.01); feeling that their identity more negatively impacted the ED experience (p<0.001); more instances of disrespect (p<0.001); and more negative feelings about their ED experience (p<0.001) (Table 2). Majority of participants in both groups shared ED experiences focusing on healthcare providers (p = 0.6) and an experience occurred in the 6 months prior to survey completion date (p = 0.2) (Table 2).

### Quantitative and qualitative findings

Results of statistically significant triad and dyad analyses are presented along with the results of the thematic analysis to contextualize findings. Overall, the qualitative analysis of shared micronarratives among YMHC identified four major themes: *Judgement and stigmatization; Unmet care needs; MH expertise;* and *ED environment.*

**Judgement and stigmatization.** Fig 2 presents a triad (T1) demonstrating that YMHC were statistically more likely to report feelings of judgment than youth controls. Fig 2 also highlights a dyad (S2) demonstrating that YMHC were statistically more likely to share experiences about ED staff than hospital processes compared to youth controls (p = 0.01).

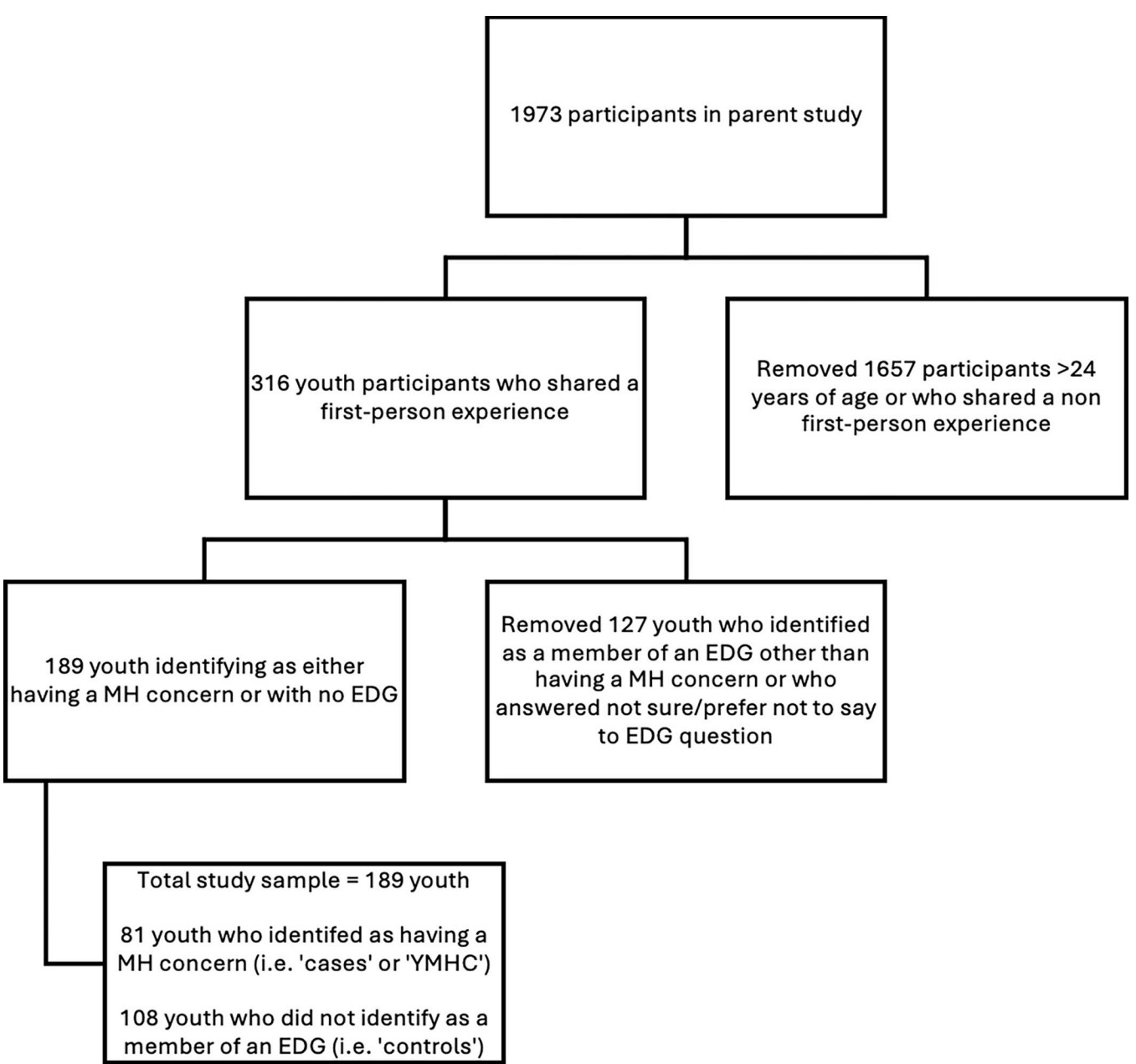

**Fig 1. Flow diagram depicting identification of study sample.**

Qualitative analysis revealed that most YMHC had negative interactions with ED staff. YMHC described feelings of judgement/stigmatization and either fear of, or perceived, bias and negative treatment by staff for having visible self-harm or history of MH concerns.

"Whenever I was approached by a nurse or staff, security officers were there as if I would put them in danger. I went in seeking for some support and only thing I was left with is an empty heart and neglected feeling."

- Youth who visited the ED following a suicide attempt with visible self-harm

**Table 1. Demographic characteristics of study participants [a].**

| Variable | Total (N = 189) n (%) | Youth with Mental Health Concerns | | $X^2$ N (df) | P-value |
| --- | --- | --- | --- | --- | --- |
| | | Yes (n = 81) n (%) | No (n = 108) n (%) | | |
| **Gender identity*** | | | | | |
| Woman | 117 (61.9) | 52 (64.2) | 65 (60.2) | 7.2 (3) | **0.05** |
| Man | 58 (30.7) | 19 (23.4) | 39 (36.1) | | |
| Non-binary | 11 (5.8) | 8 (9.9) | 3 (2.8) | | |
| Not sure/Prefer not to say | 3 (1.6) | 2 (2.5) | 1 (0.9) | | |
| **Gender Diverse*** | | | | | |
| Yes | 18 (9.5) | 15 (18.5) | 3 (2.8) | 14.4 (2) | < .001 |
| No | 162 (85.7) | 61 (75.3) | 101 (93.5) | | |
| Not sure/Prefer not to say | 9 (4.8) | 5 (6.2) | 4 (3.7) | | |
| **Frequency of "struggling to make ends meet" *** | | | | | |
| Never | 56 (29.6) | 15 (18.5) | 41 (38.0) | 19.1 (5) | < .001 |
| Rarely | 34 (18.0) | 12 (14.8) | 22 (20.4) | | |
| Sometimes | 40 (21.2) | 19 (23.4) | 21 (19.4) | | |
| Often | 18 (9.5) | 11 (13.6) | 7 (6.5) | | |
| All the time | 21 (11.1) | 16 (19.8) | 5 (4.6) | | |
| Not sure/prefer not to say | 20 (10.6) | 8 (9.9) | 12 (11.1) | | |
| **Ethnicity Identity [b, c]** | | | | | |
| White/European | 105 (55.6) | 42 (51.9) | 63 (58.3) | 1.2 (3) | 0.5 |
| Indigenous | 7 (3.7) | 4 (4.9) | 3 (2.8) | | |
| Black | 2 (1.1) | 0 (0.0) | 2 (1.85) | | |
| Other (Latin American, South, Southeast, or West Asian, Filipino, other) | 13 (6.9) | 5 (6.2) | 8 (7.4) | | |
| One or more ethnicity | 1 (0.5) | 1 (1.2) | 0 (0.0) | | |
| Not sure/prefer not to say | 5 (2.6) | 3 (3.7) | 2 (1.85) | | |
| Missing data | 56 (29.6) | 26 (32.1) | 30 (27.8) | | |
| **Identification with additional EDGs among youth with MH concerns** | - | **(n = 81)** | - | - | - |
| No additional groups | - | 14 (17.3) | - | - | - |
| 1 additional group | | 26 (32.1) | | | |
| 2 additional groups | | 41 (50.6) | | | |

*Statistically significant (p<0.05)

[a] P-values do not include those with missing data, including those who answer not sure/prefer not to say which were recoded as missing data for the purposes of the analysis.

[b] Variables were collapsed from original survey response options due to small cell sizes.

[c] Fisher exact statistic reported because >20% of cells had a cell count of <5 [31].

Another youth who identified as a vulnerably housed woman stated:

"I went [. . .] to get some self-harm cuts looked at cause they were infected. The doctor prejudicially asked if I was trying to kill myself, then told me the cuts that were clearly infected, were seemingly "not infected". Every time I go to the [ED], they visibly treat me differently if I'm in there for mental health or addiction issues."

Qualitative analysis also revealed some positive emotions when interacting with staff including feeling "comfortable" and "safe".

FGD participants echoed having largely negative experiences with ED staff and that perceived prejudice was common when presenting to the ED with a known history of substance use or MH concerns.

**Table 2. Characteristics of ED visits [a].**

| Variable | Total (N = 189) n (%) | Youth with Mental Health Concerns | | $X^2$ N (df) | P-value |
| --- | --- | --- | --- | --- | --- |
| | | Yes (n = 81) n (%) | No (n = 108) n (%) | | |
| **ED visit frequency in last 2 years prior to the story shared [b]** | | | | | |
| 0 | 29 (15.3) | 9 (11.1) | 20 (18.5) | 11.6 (4) | < .01 |
| 1–3 | 57 (30.2) | 20 (24.7) | 37 (34.3) | | |
| ≥ 4 times | 27 (14.3) | 19 (23.5) | 8 (7.4) | | |
| Not sure/prefer not to say | 19 (9.5) | 7 (8.6) | 11 (10.2) | | |
| Missing data | 58 (30.7) | 26 (32.1) | 32 (29.6) | | |
| **Interval between ED visit and date of survey completion [a]** | | | | | |
| 0–6 months | 70 (37.0) | 32 (39.5) | 38 (35.2) | 6.6 (4) | 0.2 |
| 7–120 months | 23 (12.2) | 12 (14.8) | 11 (10.2) | | |
| >12 months | 36 (19.1) | 11 (13.6) | 25 (23.1) | | |
| Not sure/prefer not to say | 4 (2.1) | 0 (0.0) | 4 (3.7) | | |
| Missing data | 56 (29.6) | 26 (32.1) | 30 (27.8) | | |
| **Focus of Experience [b, c]** | | | | | |
| Healthcare providers (doctors, nursing staff) | 64 (33.9) | 25 (30.9) | 39 (36.1) | 3.0 (5) | 0.6 |
| Waiting room | 25 (13.2) | 8 (9.9) | 17 (15.7) | | |
| Other hospital areas (triage, registration, discharge) | 13 (6.9) | 6 (7.4) | 7 (6.5) | | |
| Other hospital staff (social workers, security officers, porters, and imaging technicians) | 11 (5.8) | 6 (7.4) | 5 (4.6) | | |
| Other | 5 (2.7) | 3 (3.7) | 2 (1.9) | | |
| Not sure/prefer not to say | 15 (7.9) | 7 (8.6) | 8 (7.4) | | |
| Missing data | 56 (29.6) | 26 (32.1) | 30 (27.8) | | |
| **Effect of identity on experience* [b]** | | | | | |
| Negative | 34 (18.0) | 30 (37.1) | 4 (3.7) | 35.9 (3) | < .001 |
| No effect | 114 (60.3) | 37 (45.7) | 77 (71.3) | | |
| Positive | 16 (8.5) | 7 (8.6) | 9 (8.3) | | |
| Not sure/prefer not to say | 25 (13.2) | 7 (8.6) | 18 (16.7) | | |
| **Experience about being treated without respect*** | | | | | |
| Yes | 49 (25.9) | 33 (40.7) | 16 (14.8) | 16.8 (2) | < .001 |
| No | 116 (61.4) | 38 (46.9) | 78 (72.2) | | |
| Not sure/prefer not to say | 24 (12.7) | 10 (12.4) | 14 (13.0) | | |
| **Overall feelings about experience at time it occurred*** | | | | | |
| Positive (feeling accepted, happy, hopeful, relieved, satisfied, or thankful | 70 (37.0) | 20 (24.7) | 50 (46.3) | 20.6 (3) | < .001 |
| Negative (feeling afraid, disappointed, embarrassed, frustrated, helpless, or worried) | 83 (43.9) | 49 (60.5) | 34 (31.5) | | |
| Mixed | 24 (12.7) | 11 (13.6) | 13 (12.0) | | |
| Missing data | 12 (6.4) | 1 (1.2) | 11 (10.2) | | |

*Statistically significant (p≤0.05)

[a] P-values do not include those with missing data, including those who answer not sure/prefer not to say which were recoded as missing data for the purposes of the analysis.

[b] Variables were collapsed from original survey response options due to small cell sizes and to improve reader clarity in a meaningful and evidence-based way.

[c] Fisher exact statistic reported because >20% of cells had a cell count of <5 [31]

**Unmet care needs.** Fig 3 highlights a triad (T5) showing that YMHC were statistically more likely to report feeling unsure and unsupported in coping with their concern than controls. Fig 3 also demonstrates a dyad (S4) showing YMHC were statistically less likely to report having enough attention paid to their needs than controls (p = 0.001).

**T1. During the events in the shared story, the patient was...*****

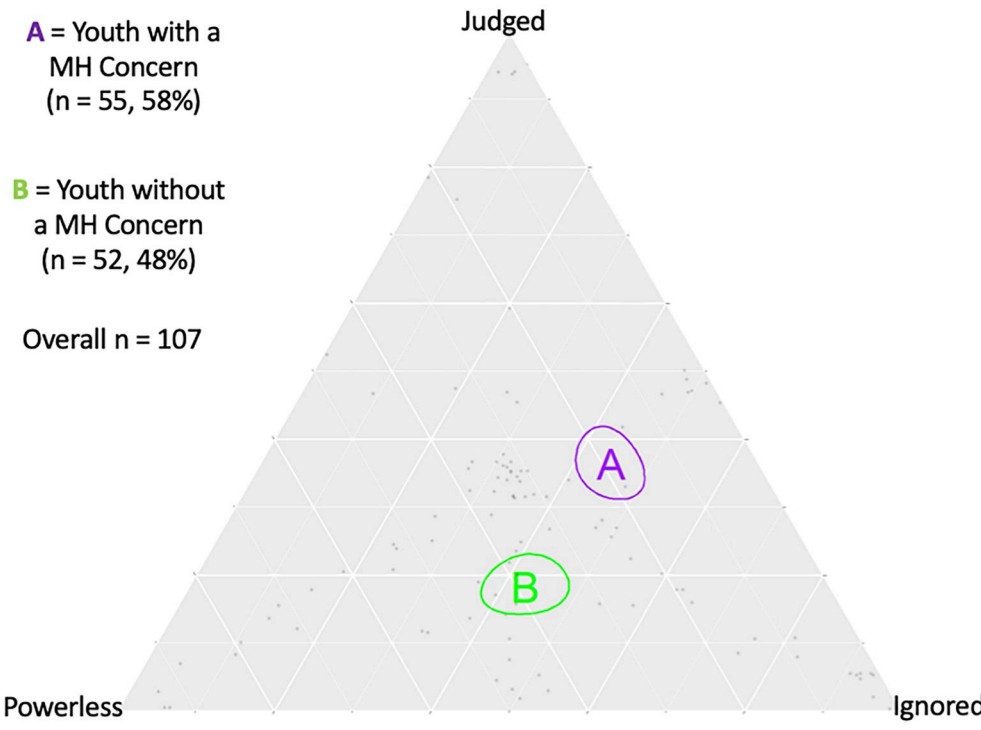

**S2. The events in the story were mostly about... *****

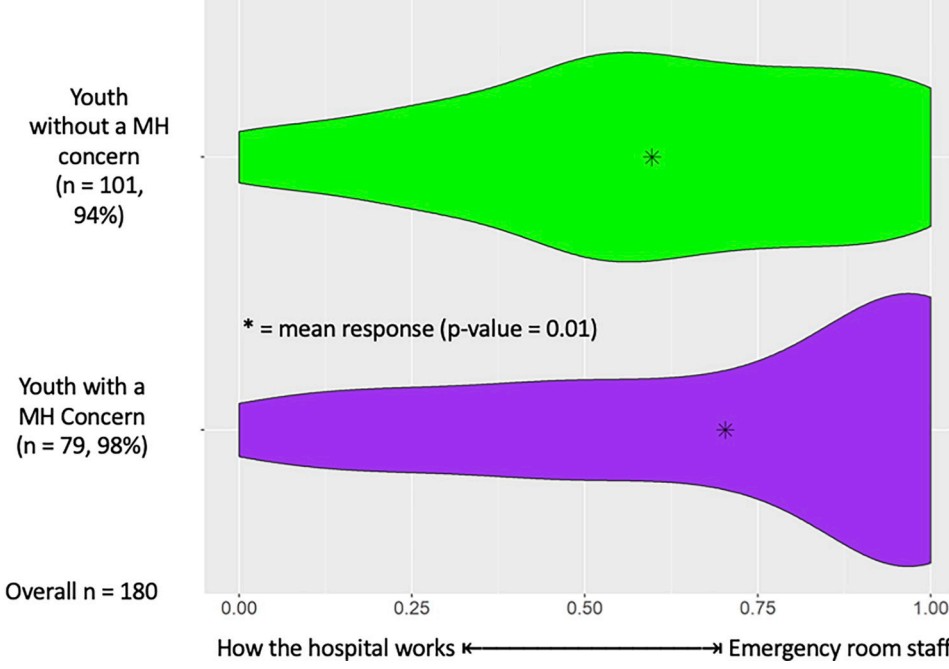

**Fig 2. Triad (T1) regarding participants' negative feelings and dyad (S2) regarding topic of ED experience shared.**
The number of responses from each group and associated percent of total group sample are provided, along with total
number of responses, with *** indicating statistical significance (p<0.05).

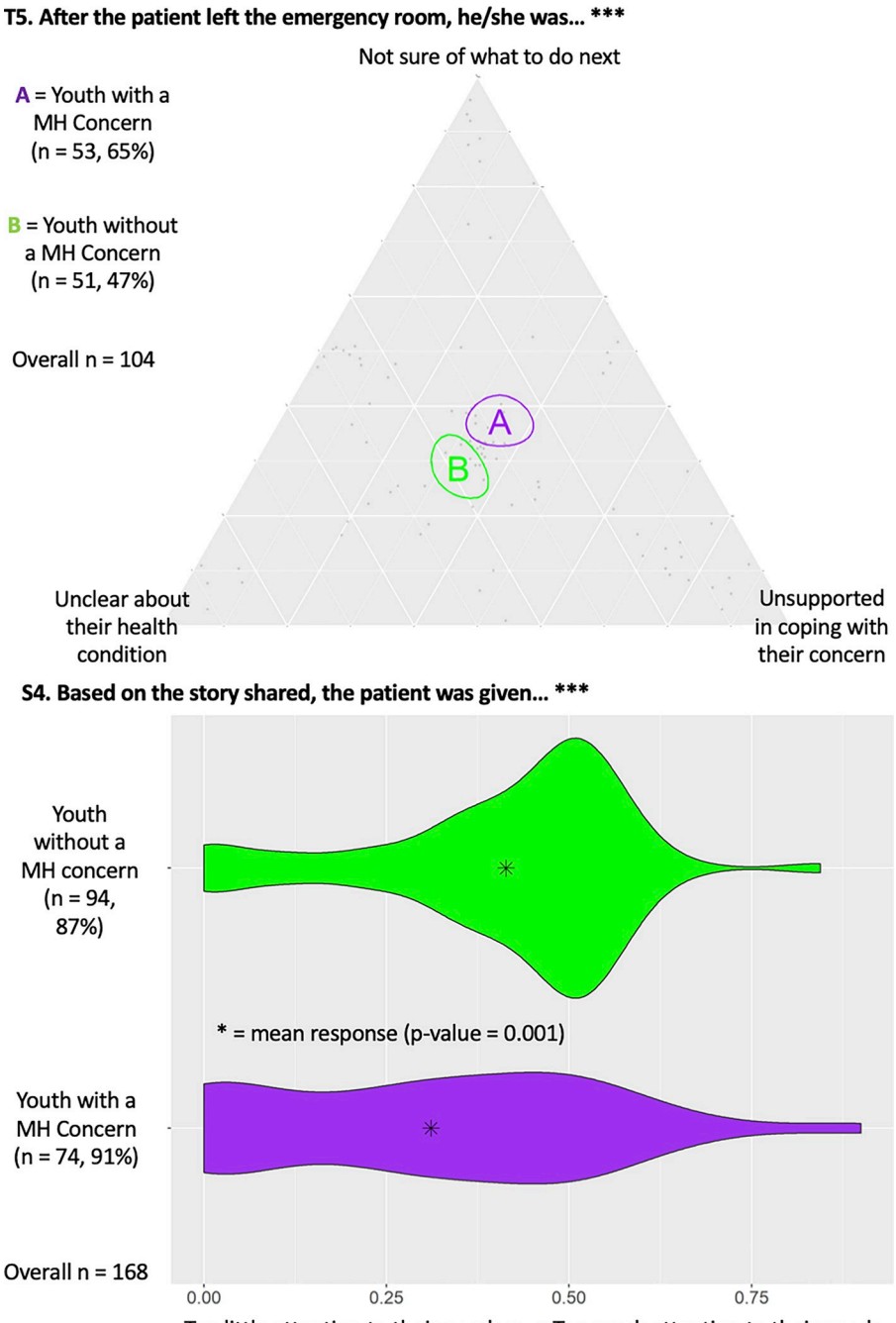

**Fig 3. Triad (T5) and dyad (S4) regarding attention to participant needs in the ED among YMHC.** The number of responses from each group and associated percent of total group sample are provided, along with total number of responses, with *** indicating statistical significance (p<0.05).

Thematic analysis revealed that the majority of youth reported unmet physical and MH care needs in the ED. YMHC often reported feeling *'unheard'*, invalidated, and that the MH and physical health symptom severity was underestimated, under investigated, and inadequately addressed because of having a known MH condition, a concept known as diagnostic overshadowing [32].

"[. . .] I've been in some life threating situations and been overlooked or passed through triage and waiting for 6 hours, or a simple sprained wrist will cause an admittance to [the mental health ward]"

- Youth who identified as having a bipolar disorder

The intersection of having a MH concern with other EDGs such as identifying as LGBTQ2S+ was also reported as contributing to diagnostic overshadowing. For example, a youth who identified as gender-diverse stated:

"[. . .] Identifying as transgender is terrifying in this hospital, my validity as a person/patient is constantly questioned, as well as the severity of my symptoms."

Positive experiences among YMHC were reported when their issues were taken seriously, adequately investigated, and they felt validated by staff for coming to the hospital.

FGD participants agreed with themes relating to diagnostic overshadowing in the ED, particularly as it relates the intersection of MH concerns with substance use and unstable housing. FGD participants endorsed that their care needs were frequently left unmet and felt they were often discharged too early, especially when presenting with self-harm or suicide concerns in the ED.

**MH expertise.** Fig 4 illustrates a triad (T6) demonstrating that, compared to controls, YMHC were statistically more likely to report that a better understanding of personal situation, identity and culture would improve future ED care. Fig 4 also highlights results from a dyad (S1) showing that YMHC were statistically less likely to report receiving enough attention to their personal situation, identity or culture compared to youth controls (p = 0.004).

Qualitative analysis revealed that YMHC perceived lack of MH expertise and knowledge, of working with YMHC in the ED. Participants reported negative experiences when healthcare providers did not consider or acknowledge their MH identity or trauma history during interactions in the ED.

"[I] went to the ER for faintness and dizziness. I have a lot of anxiety and I feel like that wasn't given much attention."

- Youth who identified as a woman

Continuity of ED care with other MH services contributed to positive experiences among YMHC. During ED visits the experience was perceived by some YMHC as addressing gaps in community MH care and facilitating connections with MH experts in the ED.

"[. . .] I came here and they helped me get a psychologist [. . .] and put me in the hospital for five days. And it was good. I am happy."

- Youth who identified as a man with a physical disability

During FGDs, participants perceived that some healthcare providers have the education on MH concerns, but do not engage in providing the appropriate support. FGD participants also shared positive experiences when they were connected to psychiatric services and received referrals to MH, substance use, and housing programs.

**ED environment.** Fig 5 illustrates a triad (T4) depicting that YMHC were statistically more likely to report having their experience affected by staff behaviour compared to youth controls. Fig 5 also displays a dyad (S5) exhibiting that YMHC were statistically more likely to

**T6. Based on the shared story, the following would improve future emergency room care...**
**\*\*\***

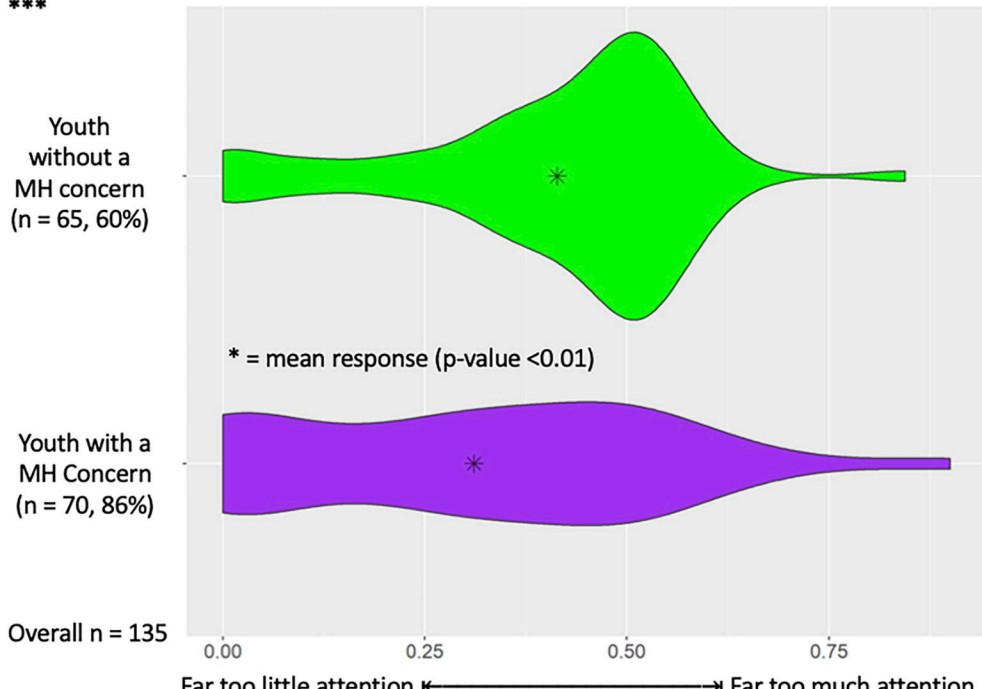

**A** = Youth with a MH Concern (n = 44, 54%)

**B** = Youth without a MH Concern (n = 56, 52%)

Overall n = 100

Better understanding of personal situation, identity and culture

Easier access to medical care

Better communication between healthcare providers

**S1. During the ED visit, the patient's personal situation, identity or culture received...**
**\*\*\***

Youth without a MH concern (n = 65, 60%)

\* = mean response (p-value <0.01)

Youth with a MH Concern (n = 70, 86%)

Overall n = 135

Far too little attention ↤————————————↦ Far too much attention

**Fig 4. Triad (T6) showcasing focus of improvement for future ED care and dyad (S1) showing attention paid to participant's personal situation, identity, or culture.** The number of responses from each group and associated percent of total group sample are provided, along with total number of responses, with \*\*\* indicating statistical significance (p<0.05).

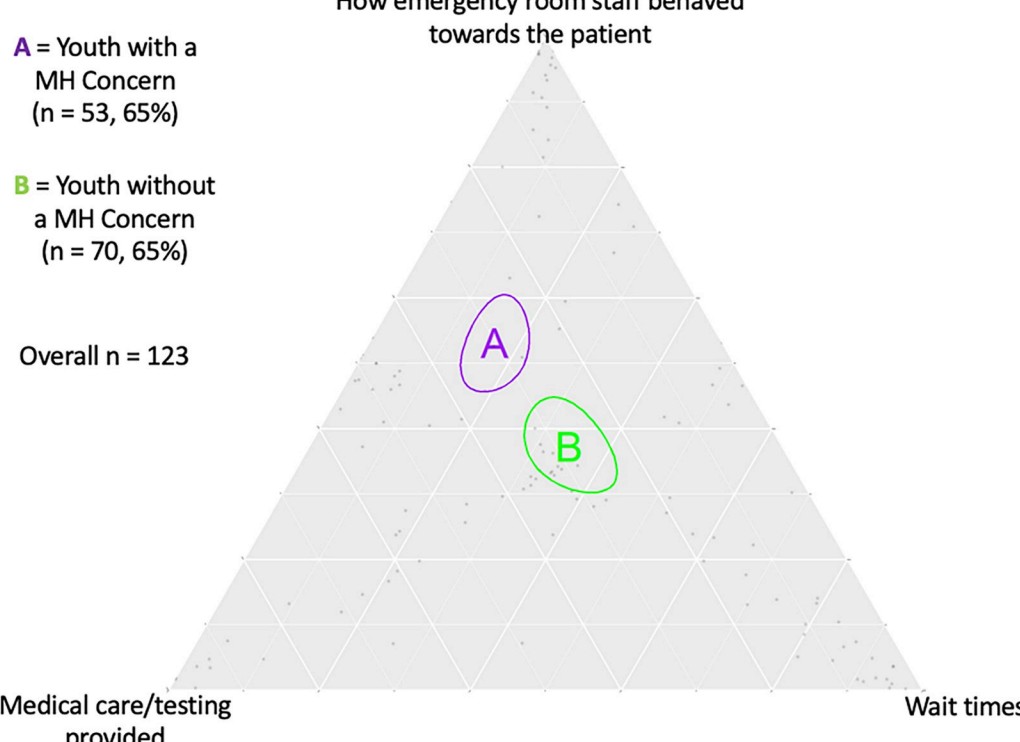

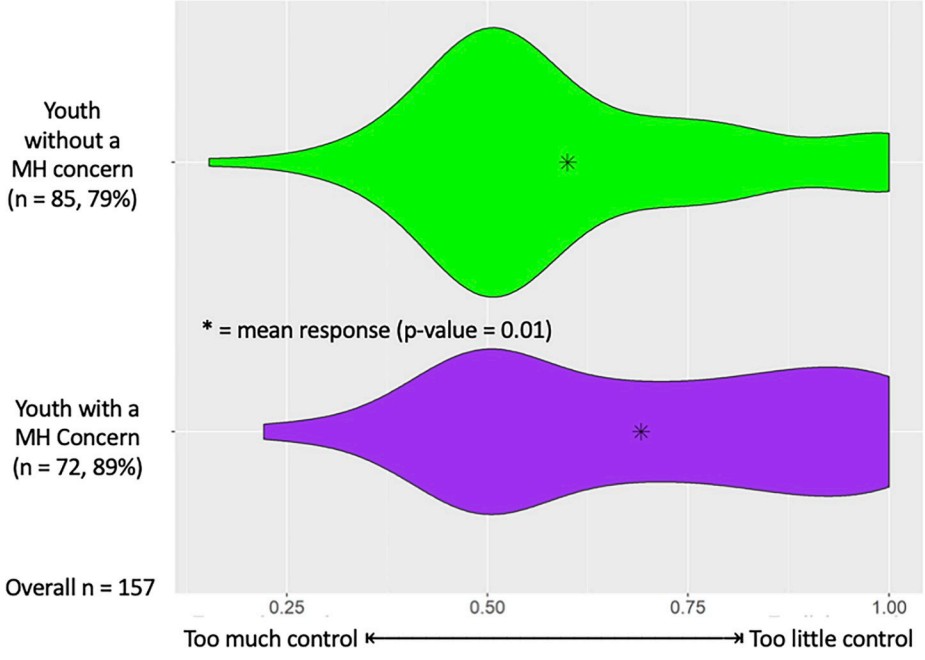

**Fig 5. Triad (T6) showcasing focus of improvement for future ED care and dyad (S5) showing feelings of control about care decisions.** The number of responses from each group and associated percent of total group sample are provided, along with total number of responses, with *** indicating statistical significance (p<0.05).

report having too little control in making decisions about their care compared to controls (p = 0.01).

Lengthy wait times and being moved to different rooms were reported as exacerbating MH concerns during ED visits, and experiences were reported as more positive by YMHC if wait times were shorter.

"Truthfully, I hate coming to the hospital [..] The wait times are always extremely long. I'm often left just standing or sitting in hallways. [. . .] It's just a very long process and I've never felt it's been worth it."

- Youth participant who identified as a woman

Processes relating to restraint use and interactions with security personnel for management of MH symptoms in the ED were described as negative by YMHC, including comments about physical harm, clothing indecency and lack of communication regarding restraints. Additionally, policies enforced during the COVID-19 pandemic requiring individuals to come alone to the hospital without parents or guardians exacerbated MH symptoms.

"Because of Covid, I had to go into the building alone, but because I was very mentally unwell, I was having trouble communicating with people and understanding what to do, and in turn made me more confused and erratic."

- Youth who identified as non-binary

FGD participants also endorsed feeling "*hypersensitive*" to the ED environment with long-waits and transfer throughout the ED was perceived as stressful, and noted that it can be particularly for people with a history of a MH concern, trauma, or vulnerable housing. FGD participants also reported negative experiences with physical restraints, and proposed having peer supports in the ED as a possible solution.

## Discussion

Study findings demonstrate that YMHC have more negative ED care experiences compared to age-matched youth who do not identify as equity-deserving. YMHC experienced negative staff attitudes and treatment and perceived stigmatization. This is consistent with previous literature demonstrating that people with MH concerns experience diagnostic overshadowing for visible self-harm or having MH labels, and report experiencing less than usual care and treatment because of having a MH history [16, 17, 19]. YMHC also reported exacerbation of MH conditions due to wait-times and restraint use. These findings are all consistent with existing evidence among youth and adults with MH concerns [13–19]. For example, a 2022 study on young people presenting to the ED for suicide in Australia also found that aspects of the ED environment and attitudes and treatment by staff contributed to negative experiences among youth [18]. However, the reported negative interpersonal interactions and experiences with wait times are likely largely part of systemic issues contributing to ED staff burnout and that may be perceived as lack of compassion and empathy by YMHC [13]. Understanding ED staff perspectives and addressing systemic barriers is therefore crucial before developing interpersonal improvement initiatives [33, 34].

Our finding that YMHC had largely unmet care needs in the ED, and the value placed on MH expertise and connection with MH experts and community services is also consistent with previous literature [10, 13, 20, 21]. This could point to the importance of trauma-informed care (TIC) when caring for MH patients, and ensuring providers are aware of TIC

concepts as they relate to MH to improve care experiences [6]. TIC involves acknowledging the impact and signs of trauma in an individual, and integrating this knowledge to provide appropriate care for an individual that avoids re-traumatization. There are a set of core principles that are integral to providing both individual and organizational TIC, which include safety, trustworthiness and transparency, peer support, collaboration, empowerment, and humility and responsiveness [35]. A 2023 Canadian study showed that youth MH patients in a pediatric ED reported better experiences when they received more help and were evaluated by MH experts [21]. These findings highlight the value of validation, interpersonal interactions, and connection to MH experts and community supports for YMHC [21]. The results of this study interpreted together with the current literature may call into question if the general ED is the best place to address MH concerns among youth, or if other community, primary care, crisis, or specialized ED supports may be more appropriate.

A unique finding in this study was the high degree of which Canadian YMH in this study also identified as members of the LGBTQ2S+ community, and the impact this had on ED experience. The finding that YMHC identified more as gender diverse than age-matched controls was expected given controls did not identify as any EDG. However, the many YMHC in our study who also identified as LGBTQ2S, and the negative impact this had on ED experience, is important. Other literature demonstrates that youth who identify as LGBTQ2S+ are at an increased risk for MH concerns, most notably youth who identify as transgender, non-binary, or gender-diverse [33, 36, 37]. However, the interaction between gender identity and seeking ED care for MH concerns has not been adequately examined [21]. Additionally, considering the intersection of having a MH concern with other EDG identities is important given the important and complex interplay of MH and other health inequities [8]. These findings, along with other findings such as the impact of COVID-19 policies on accompaniment to the ED, could suggest avenues to improve ED care experiences and reduce health inequities among YMHC.

Based on our findings and current literature, we present several strategies to improve ED care experiences for YMHC.

1. Increased presence of staff members or peer support workers with pre-existing expertise.

2. As suggested by FGD participants, increasing the presence of individuals with pre-existing MH expertise/training or lived experience could improve ED care and interactions [38].

3. ED staff education on TIC and youth-specific needs.

4. Providing education on TIC has been shown to improve understanding of MH conditions (39). Therefore, providing training on MH, caring for gender-diverse populations, and age-specific support needs could reduce misconceptions, judgement, and stigmatization when caring for youth [6, 17, 33, 34, 39, 40].

5. Improve the linkages between the ED and community MH resources.

Ensuring ED staff are aware of resources available in the community and investing in tools that ED staff could use, such as implementing follow-up telephone contacts to connect youth to outpatient treatments, has been shown to be effective and could be another route for improvement [20, 41].

## Strengths and limitations

There are a few noteworthy limitations of this research. First, it is susceptible to selection bias, as individuals not accessing care at the time of data collection may not be represented, and

recall bias, as participants were asked to share a past ED experience. Second, missing data due to a random error relating to a software update issue with the tablets used to collect survey data reduced the sample size and thus statistical power for some analyses. Third, participants were not able to indicate what MH concern(s) they identified with in the initial survey questions, so disaggregated analysis by MH concern was not feasible with this analysis. Fourth, it was not possible to identify if participants had completed the survey before, and the sample may therefore include the same participant multiple times. Fifth, this sample identified largely as white/European, was only available to English-speaking participants, and was a single-centre study, thereby limiting generalizability of results to more ethnically diverse populations, non- English speaking individuals, and other healthcare settings.

This research has numerous strengths. It is among the first to present the ED care experiences of a large sample of YMHC compared with an age-matched control group. The mixed-methods approach allows quantitative findings to be contextualized and complemented by participants' qualitative micronarratives. We made intentional efforts to ensure methodologic rigour by triangulating the results through descriptive and inferential analysis, quantitative analysis of self-interpretation questions, and thematic analysis of micronarratives, double coding on a portion of micronarratives, and constant comparison with the existing evidence base. Furthermore, the sharing of results with community members with MH concerns and experts in the field further substantiates findings. Most importantly, the use of a control group contributes to identifying experiences unique to YMHC. Finally, this paper also considers the intersection of multiple EDGs and having MH concerns, and the impact this has on ED care experiences.

## Conclusions

In summary, this study used a mixed-methods approach to expand on previous findings that YMHC have more negative ED care experiences by comparing YMHC with an age-matched control group. Future research should aim to capture ED care experiences by MH diagnosis, include more ethnically and culturally diverse populations, and seek to understand ED staff experiences and systemic barriers to care to develop meaningful interventions to improve ED care experiences for YMHC. Efforts to improve ED experiences could also be coupled with better resourcing community MH organizations and considering the most appropriate setting to address MH concerns among youth.

## Supporting information

**S1 Table. Self-interpretation survey question list.** Full list of self-interpretation dyad and triad questions with possible responses, all of which were optional to respond to.
(PDF)

**S2 Table. Multiple-choice survey question list.** Full list of survey multiple-choice questions with possible responses.
(PDF)

**S3 Table. COREQ checklist.** Completed COREQ (COnsolidated criteria for REporting Qualitative research) Checklist.
(PDF)

**S1 Dataset. Quantitative dataset.** Dataset for study sample with all quantitative data analyzed.
(XLSX)

**S1 Fig. Self-interpretation question examples.** Example of triad (left) and dyad (right) self-interpretation questions from survey through Spryng.io (23).
(PDF)

**S2 Fig. Equity deserving identity survey question.** Excerpt of survey question used to identify study comparison groups.
(PDF)

## Acknowledgments

We want to acknowledge that Queen's University is situated on traditional Haudenosaunee and Anishinaabe territory. We are grateful to be able to live, learn and play on these lands. The authors are incredibly grateful to all of the participants for sharing their experiences for this research. The authors would like to acknowledge and thank Dave McRae at the Maltby Centre for his review of the manuscript and validation of findings based on his experiences as a child and youth counsellor; Teely Hopkin, Jodi-Mae John, and Nicole Morris at Queen's University for their work developing the master codebook used throughout the qualitative thematic analysis; Patrick Norman at the Kingston General Hospital Research Institute for his assistance with running statistical analyses; Laurie Webster with Queen's Emergency Department Insight for her support with the quantitative data analysis; Reyana Jayawardena for her support facilitating focus group discussions; and the Kingston Health Sciences Centre and various community partner organizations for their involvement in this research.

## Author Contributions

**Conceptualization:** Susan A. Bartels, Melanie Walker.

**Data curation:** Susan A. Bartels, Melanie Walker.

**Formal analysis:** Laura K. Wells.

**Investigation:** Susan A. Bartels, Melanie Walker.

**Methodology:** Susan A. Bartels, Melanie Walker.

**Project administration:** Laura K. Wells.

**Resources:** Susan A. Bartels, Melanie Walker.

**Supervision:** Susan A. Bartels, Melanie Walker.

**Validation:** Laura K. Wells, Susan A. Bartels, Tania Nicholls, Melanie Walker.

**Visualization:** Laura K. Wells.

**Writing – original draft:** Laura K. Wells.

**Writing – review & editing:** Laura K. Wells, Susan A. Bartels, Tania Nicholls, Melanie Walker.

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
