## [Decision Letter · Decision Letter 0]

20 Aug 2024

PMEN-D-24-00298

Emergency department care experiences among youth with mental health concerns

PLOS Mental Health

Dear Dr. Walker,

Thank you for submitting your manuscript to PLOS Mental Health. After careful consideration, we feel that it has merit but does not fully meet PLOS Mental Health’s publication criteria as it currently stands. Therefore, we invite you to submit a revised version of the manuscript that addresses the points raised during the review process.

EDITOR: Please insert comments here and delete this placeholder text when finished. Be sure to:

Indicate which changes you require for acceptance versus which changes you recommendAddress any conflicts between the reviews so that it's clear which advice the authors should followProvide specific feedback from your evaluation of the manuscript

Please ensure that your decision is justified on PLOS Mental Health’s publication criteria and not, for example, on novelty or perceived impact.

We look forward to receiving your revised manuscript.

Kind regards,

Rachel Dale, PhD

Academic Editor

PLOS Mental Health

Journal Requirements:

Additional Editor Comments (if provided):

Thanks to the authors for this paper. The reviewers and I suggest some minor revisions prior to publication.

Please pay particular attention to the reviewer’s comments regarding the use if abbreviations and the term ‘equity-deserving’.

In general, there is an extensive use of abbreviations that can make the manuscript a bit difficult to follow. For example on line 76, to what does EM refer? Additionally line 102, KHSC ED/UCC is a new abbreviation and since it is likely only used rarely, can perhaps be written out in full. Line 219: HCP?

Lines 200-201: here both sex and gender are referred to. Therefore I just want to check that in the focus groups participants were categorised according to sex and in the survey they were categorised according to gender? If not then this should be corrected such that female/male/intersex refer to sex and man/woman/gender-diverse refer to gender.

Results: please report full statistics rather than just the p-values.

Table 2: how is it possible that 29 participants had no ED visit in the prior 2 years when on line 106 it states that this is part of the inclusion criteria? As I understand it the purpose of this study is to assess people with direct ED visit experience (lines 124-125) so these 29 participants should have been excluded.

Minor points:

Line 69-70: some commas are missing - and improve YMHC experiences with, and interaction with, the health system,…

Line 140: The Chi-squared test is an inferential statistic.

Line 365: for suicidal behaviour? suicidal ideation?

Line 375: perhaps trauma-informed care could be very briefly explained.

Reviewers' comments:

Reviewer's Responses to Questions

**Comments to the Author**

1. Does this manuscript meet PLOS Mental Health’s publication criteria? Is the manuscript technically sound, and do the data support the conclusions? The manuscript must describe methodologically and ethically rigorous research with conclusions that are appropriately drawn based on the data presented.

Reviewer #1: Yes

Reviewer #2: Yes

2. Has the statistical analysis been performed appropriately and rigorously?

Reviewer #1: Yes

Reviewer #2: Yes

3. Have the authors made all data underlying the findings in their manuscript fully available (please refer to the Data Availability Statement at the start of the manuscript PDF file)?

Reviewer #1: Yes

Reviewer #2: Yes

4. Is the manuscript presented in an intelligible fashion and written in standard English?

Reviewer #1: Yes

Reviewer #2: Yes

5. Review Comments to the Author

Reviewer #1: Thanks for the opportunity to review this paper. It adds to existing evidence regarding the need for improving ED support for young people with mental health problems.

There are some points which require amendment or clarification before publication:

Abstract:

- what do you mean by 'equity deserving'? - suggest leaving this terminology out if there isn't space to define it

- can you add something to the methods section about how you collected qualitative data? It's not clear if this was interviews, a qual survey etc

Introduction:

- line 83 - The last sentence of your introduction doesn't make sense

- in general I think the rationale for why the present research is needed could be made stronger. You have presented lots of previous research in this area already - why exactly is your study needed and how does it build on what has gone before?

Methods:

- is it correct that young people didn't need to have visited the ED with a concern relating to their mental health to be in the EDG group? please make this explicit in the text if so.

- how are you defining .'micronarratives'?

- your primary outcome of 'ED care experiences' isn't really an outcome - can you make this more specific?

- you should add references for the steps in your qualitative analysis

Results:

- can you add some information as participant identifiers after the quotes? This might read better than introducing each quote by participant characteristics.

Reviewer #2: The study is well presented and is very interesting for the mixed methodology used and its most relevant findings as well as for the proposals for improvement.

Please review the use of the abbreviation MH, so that it is explained the first time it is used.

Please explain why you included in the sample people who had no ED visits in the last 2 years (table2).

6. PLOS authors have the option to publish the peer review history of their article (what does this mean?). If published, this will include your full peer review and any attached files.

**Do you want your identity to be public for this peer review?** For information about this choice, including consent withdrawal, please see our Privacy Policy.

Reviewer #1: **Yes: **Dr Rebecca Appleton

Reviewer #2: **Yes: **Vania Martínez

---

## [Decision Letter · Decision Letter 1]

5 Nov 2024

PMEN-D-24-00298R1

Emergency department care experiences among youth with mental health concerns

PLOS Mental Health

Dear Dr. Walker,

Thank you for submitting your manuscript to PLOS Mental Health. After careful consideration, we feel that it has merit but does not fully meet PLOS Mental Health’s publication criteria as it currently stands. Therefore, we invite you to submit a revised version of the manuscript that addresses the points raised during the review process.

The reviewers and I agree the revision has addressed all comments. Thanks to the authors for their responses!

As a final minor edit, it has come to my attention that PLOS Mental Health has specific guidelines for reporting qualitative research:

PLOS Mental Health considers qualitative and mixed-methods studies for publication. We recommend that authors use the Consolidated criteria for reporting qualitative research (COREQ) checklist or Standards for reporting qualitative research (SRQR) checklist. (http://journals.plos.org/plosmentalhealth/s/submission-guidelines#loc-qualitative-research).

Please check that your reporting follows the items of one of the above checklists and state in the methods section which checklist was adhered to. Apologies for not including this in the original review. However, I believe the manuscript mostly follows the checklist criteria so I hope this will not be too much work.

We look forward to receiving your revised manuscript.

Kind regards,

Rachel Dale, PhD

Academic Editor

PLOS Mental Health

Reviewers' comments:

Reviewer's Responses to Questions

**Comments to the Author**

1. If the authors have adequately addressed your comments raised in a previous round of review and you feel that this manuscript is now acceptable for publication, you may indicate that here to bypass the “Comments to the Author” section, enter your conflict of interest statement in the “Confidential to Editor” section, and submit your "Accept" recommendation.

Reviewer #1: All comments have been addressed

2. Does this manuscript meet PLOS Mental Health’s publication criteria? Is the manuscript technically sound, and do the data support the conclusions? The manuscript must describe methodologically and ethically rigorous research with conclusions that are appropriately drawn based on the data presented.

Reviewer #1: Yes

3. Has the statistical analysis been performed appropriately and rigorously?

Reviewer #1: Yes

4. Have the authors made all data underlying the findings in their manuscript fully available (please refer to the Data Availability Statement at the start of the manuscript PDF file)?

Reviewer #1: Yes

5. Is the manuscript presented in an intelligible fashion and written in standard English?

Reviewer #1: Yes

6. Review Comments to the Author

Reviewer #1: The authors have addressed all my previous comments - no further edits required.

7. PLOS authors have the option to publish the peer review history of their article (what does this mean?). If published, this will include your full peer review and any attached files.

**Do you want your identity to be public for this peer review?** For information about this choice, including consent withdrawal, please see our Privacy Policy.

Reviewer #1: **Yes: **Dr Rebecca Appleton

---

## [Editor Report · Decision Letter 2]

19 Nov 2024

Emergency department care experiences among youth with mental health concerns

PMEN-D-24-00298R2

Dear Dr. Walker,

We are pleased to inform you that your manuscript 'Emergency department care experiences among youth with mental health concerns' has been provisionally accepted for publication in PLOS Mental Health.

Before your manuscript can be formally accepted you may need to complete some formatting changes, which you will receive in a follow up email. A member of our team will be in touch with a set of requests.

Best regards,

Rachel Dale, PhD

Academic Editor

PLOS Mental Health